# Prognostic value of copeptin in patients with acute coronary syndrome: A systematic review and meta-analysis

**Jiapeng Lu**⊙*, **Siming Wang, Guangda He, Yanping Wang**

National Clinical Research Center of Cardiovascular Diseases, State Key Laboratory of Cardiovascular Disease, Fuwai Hospital, National Center for Cardiovascular Diseases, Chinese Academy of Medical Sciences and Peking Union Medical College, Beijing, People's Republic of China

* jiapeng.lu@fwoxford.org

## Abstract

### Background

The aim of this study was to evaluate the value of copeptin in predicting mortality including both short-term and long-term mortality in patients with acute coronary syndrome (ACS).

### Methods

Potential studies were searched and selected through PubMed, Embase and Cochrane databases up to December 2019. The predictive performance was evaluated by the pooled sensitivity and specificity, and summary receiver operating characteristic curves. Cochran's Q test and $I^2$ index were used to assess between-study heterogeneity, and Deek's test and funnel plots were used to assess publication bias.

### Results

Total six studies comprising 2269 patients were included in this meta-analysis. The area under the receiver operating characteristic curve of copeptin in predicting mortality in patients with ACS was 0.73 (95% CI: 0.69–0.77). The pooled sensitivity and specificity of copeptin were 0.77 (95% CI: 0.59–0.89) and 0.60 (95% CI: 0.47–0.71), respectively. Significant between-study heterogeneity was identified in both sensitivity ($P = 0.01$; $I^2 = 69.76\%$) and specificity ($P<0.001$; $I^2 = 97.32\%$) among the six included studies. The meta-regression analysis indicated that the number of study centers was significantly associated with the heterogeneity of sensitivity ($P = 0.03$), whereas the study design ($P = 0.03$) and duration of follow-up ($P<0.001$) were significantly associated with the heterogeneity of specificity.

### Conclusions

Copeptin has acceptable prognostic value for mortality in patients with ACS. Further studies based on multimarker strategy are needed to evaluate the prognostic value of copeptin for ACS in conjunction with other well-established biomarkers.

**Data Availability Statement:** All relevant data are within the manuscript.

**Funding:** JL was supported by the National Natural Science Foundation of China (81903399) and the National Key Research and Development Program

(2017YFC1310801, 2017YFC1310803) from the Ministry of Science and Technology of China. The funders had no role in study design, data collection and analysis, decision to publish, or preparation of the manuscript.

**Competing interests:** The authors have declared that no competing interests exist.

## Introduction

Coronary artery disease (CAD) is the leading cause of death worldwide. Patients with CAD, especially acute coronary syndrome (ACS), are always at the high risk of recurrent cardiovascular events and death [1, 2]. Thus, there is a need to improve risk stratification in ACS using biomarkers beyond traditional cardiovascular risk factors to optimize treatment in clinical practice.

To date, several biomarkers such as natriuretic peptide, cardiac troponins and arginine vasopressin (AVP) have been studied as potential biomarkers for risk stratification in patients with ACS [3–6]. Among them, AVP, a nonapeptide produced in the hypothalamus, has been proved that it contributes to osmoregulation and cardiovascular homeostasis [7, 8]. However, AVP cannot be reliably measured in plasma due to its low stability. Copeptin, the C-terminal portion of provasopressin, is regarded as the ideal surrogate biomarker for AVP due to its favorable stability in blood [9, 10]. Thus, copeptin has been thought to be a potential biomarker for several acute illness, such as lower respiratory infection, acute pancreatitis, stroke and ACS [11].

Some studies have indicated that copeptin is a strong prognostic predictor on death in patients with ACS [12–17]. In contrast, no significant association of copeptin with the prognosis for survival after ACS was reported in other studies [18, 19]. The controversial findings might result from the differences in study design and limited sample size. Additionally, most of previous studies were conducted in single center, which might influence the generalizability of the results [13–16]. Accordingly, we aim to set out a systematic review and meta-analysis to evaluate the association between copeptin and mortality, and to quantify the value of copeptin in predicting mortality in patients with ACS.

## Methods

### Literature search and selection

We searched for relevant studies published in English up to December 2019 through the Pubmed/MEDLINE, EMBASE and Cochrane database with the following terms and their combinations: "copeptin", "C-terminal provasopressin", "coronary artery disease", "coronary heart disease", "angina", "myocardial infarction", "death" and "mortality".

We selected eligible studies from all the relevant literatures found in databases by orderly reviewing title, abstract and full text. The inclusion criteria were as follows: (1) studies focused on the value of copeptin in predicting mortality including both in-hospital and long-term mortality in patients with ACS or suspected ACS; (2) prospective cohort studies or randomized control trials (RCT); (3) studies with at least 1 month follow-up. Studies were excluded if: (1) we are not able to calculate sensitivity and specificity based on the data in the literature; (2) literatures are conference articles, editorial, letters, reviews or duplicated publications.

### Data extraction and quality assessment

All the following information was separately extracted by two investigators: the first author's name, year of publication, study design, number of study centers, study outcome, duration of follow-up, number of patients enrolled, cut-off value of copeptin, number of true positive (TP), number of false positive (FP), number of false negative (FN) and number of true negative (TN). Discrepancies were resolved through discussion with the third investigator.

Quality assessment was conducted using the Quality Assessment of Diagnostic Accuracy Studies-2 (QUADAS-2) [20]. The QUADAS-2 form is composed of four domains: (1) patient selection, (2) index test, (3) reference standard and (4) flow and timing. For each domain, the risk of bias and applicability concerns was analyzed and rated as "low", "high" and "unclear".

Two investigators independently assessed the quality of the eligible studies and resolved disagreements by discussion.

## Statistical analysis

We calculated pooled sensitivity and specificity using a random-effects model [21]. Forest plots of each study and pooled estimates for sensitivity and specificity with 95% confidence intervals (95% CI) were used to visualize these results. We also presented a summary receiver operating characteristic curve (SROC) and calculated the area under the curve (AUC). The AUC close to 1 indicated a good performance for predicting mortality in ACS. Additionally, between-study heterogeneity of sensitivity and specificity was analyzed using the Cochrane's Q test and $I^2$ index, and $P<0.1$ or $I^2>50\%$ indicated statistically significant heterogeneity. Meta-regression analysis was conducted to identify potential between-study heterogeneity. Deek's test was used to assess the potential publication bias, and $P<0.05$ indicated statistically significant publication bias. To assess the reliability of the pooled results, sensitivity analyses were conducted by removing one study each time to observe the influence of each study on the pooled estimates. All analyses were conducted using STATA 15.0 (STATA Corporation, College Station, Texas, USA).

## Results

### Characteristics of the eligible studies

We identified 80 relevant studies by the literature search (Fig 1). After initial review of the title and abstract, 65 studies were excluded. Among them, 18 studies were reviews, 23 studies were not designed to evaluate the prognostic effect of copeptin in patients with ACS, and 24 studies didn't focus on patients with ACS. Among the 15 studies selected for full-text review, six studies didn't have sufficient data for the meta-analysis, one study was not prospective cohort study or RCT, one study didn't focus on mortality, one study didn't test baseline level of copeptin. Therefore, six studies were finally included in our meta-analysis [13–18]. Among the six included studies, five were prospective cohort studies, two were multi-center studies, and four studies had follow-up duration ≥6 months. A total of 2269 patients with ACS were studied. The characteristics of the final included studies were shown in Table 1.

The result of quality assessment was shown in Fig 2. In general, all included studies in the meta-analysis showed good quality in methodology. In this study, we only included prospective cohort studies or RCTs. Therefore, all included studies had low risk of bias in patient selection. One study had unclear risk of bias in index test domain because the cut-off value of copeptin was not pre-specified [16]. One study was judged to be unclear risk of bias in flow and timing because only part of study patients were included in the final analysis [17]. However, none of the six eligible studies needed to be excluded from the meta-analysis due to methodological defects.

### Performance of copeptin in predicting mortality in patients with ACS

The AUC of copeptin in predicting mortality in patients with ACS was 0.73 (95% CI: 0.69–0.77) (Fig 3). The pooled sensitivity and specificity of copeptin in predicting mortality in patients with ACS was 0.77 (95% CI: 0.59–0.89) and 0.60 (95% CI: 0.47–0.71), respectively (Fig 4). The positive likelihood ratio was 1.90 (95% CI: 1.60–2.30) and the negative likelihood ratio was 0.38 (95% CI: 0.23–0.64). The diagnostic odds ratio was 5.00 (95% CI: 3.00–9.00). The sensitivity analyses of diagnostic odds ratio showed that the combined estimate was 4.76 and the estimates varied between 3.57 and 5.39, which indicated that no single study could significantly influence the combined estimates (Table 2).

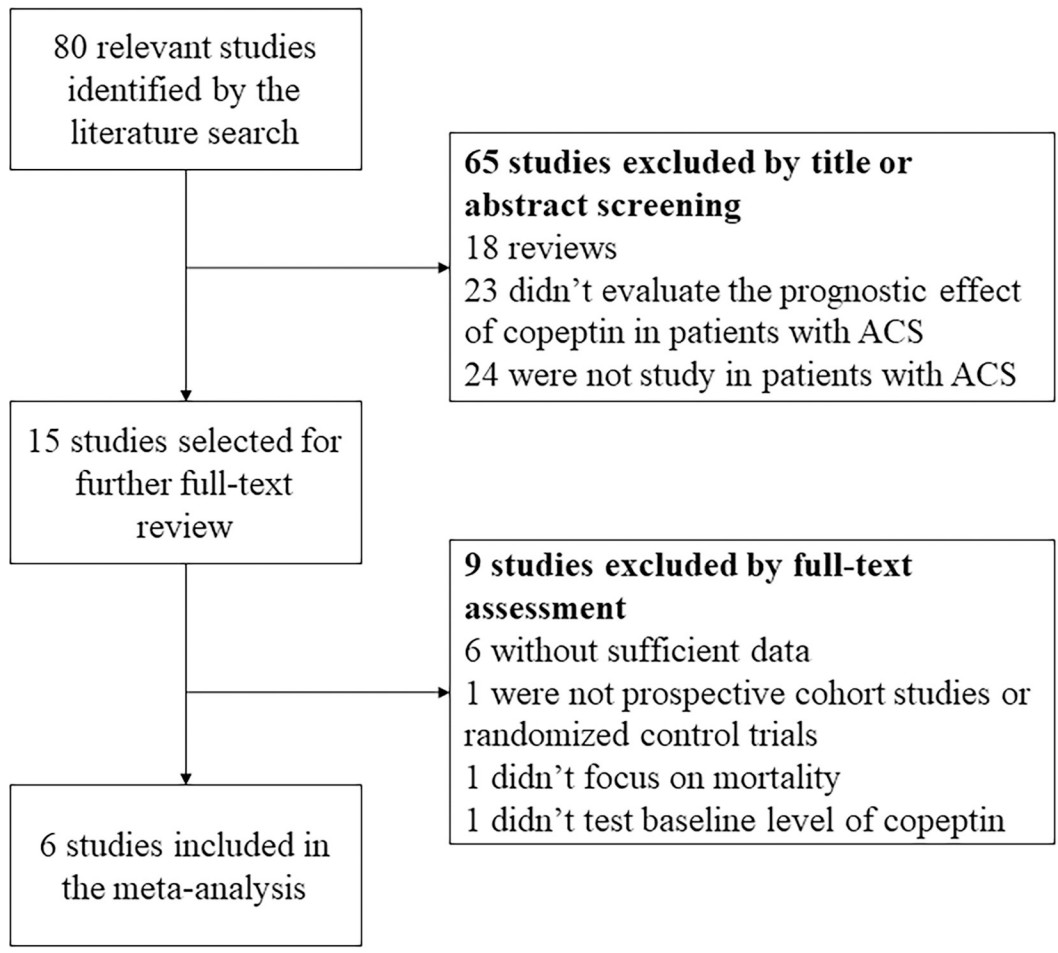

**Fig 1. Flowchart of study selection.** ACS: acute coronary syndrome.

### Between-study heterogeneity

Significant between-study heterogeneity was identified in both sensitivity ($P = 0.01$; $I^2 =$ 69.76%) and specificity ($P<0.001$; $I^2 = 97.32\%$) among the six included studies. Meta-regression was performed to examine the sources of potential heterogeneity of sensitivity and

**Table 1. Characteristics of included studies.**

| Author | Year | Study design | Sample size | No. of study center | Duration of follow-up | Study outcome | Cut-off value of copeptin (pmol/L) |
|---|---|---|---|---|---|---|---|
| Afzali, et al. [13] | 2013 | Prospective cohort study | 227 | 1 | 180 days | all-cause mortality | 14 |
| Bahrmann, et al. [14] | 2013 | Prospective cohort study | 306 | 1 | 1 year | cardiovascular mortality | 14 |
| Morawiec, et al. [15] | 2018 | Prospective cohort study | 151 | 1 | 1 year | cardiovascular mortality | 17.4 |
| Narayan, et al. [16] | 2011 | Prospective cohort study | 631 | 1 | 180 days | all-cause mortality | 7.9 |
| Sanchez, et al. [18] | 2014 | Prospective cohort study | 377 | 15 | 30 days | all-cause mortality | 25.9 |
| Vafaie, et al. [17] | 2016 | Randomized control trial | 577 | 7 | 90 days | all-cause mortality | 10 |

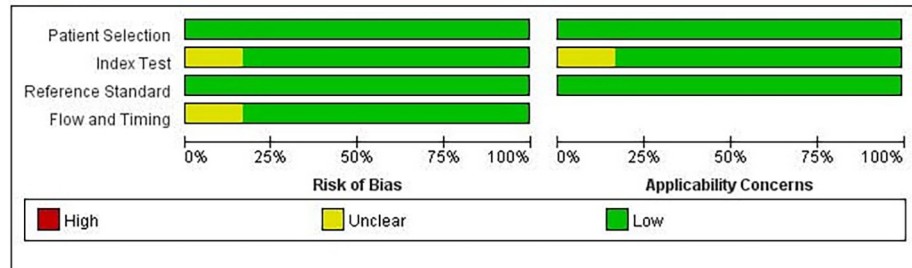

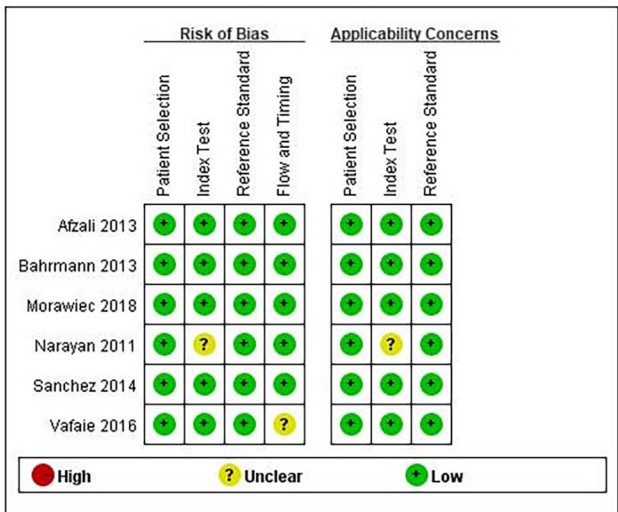

**Fig 2. Quality assessment of included studies using QUADAS-2.**

specificity (Table 3). In the meta-regression, the following covariates were included: study design (prospective cohort study or RCT), number of study centers (multicenter or single-center), sample size (<500 or ≥500), average age (<70 years or ≥70 years), duration of follow-up (<180 days or ≥180 days), study outcome (all-cause mortality or cardiovascular mortality) and cut-off value of copeptin (pre-specified or not pre-specified). The results showed that number of study centers was significantly associated with the heterogeneity of sensitivity ($P = 0.03$). Studies conducted in multiple centers (0.54, 95%: 0.23–0.85) had lower pooled sensitivity compared with studies with single center (0.85, 95%: 0.75–0.94). The study design ($P = 0.03$) and duration of follow-up ($P<0.001$) were significantly associated with the heterogeneity of specificity. The pooled specificity in clinical trial (0.78, 95%: 0.62–0.94) was much higher than that in prospective cohort studies (0.55, 95%: 0.45–0.66), and the pooled specificity obtained from studies with ≥180 days follow-up (0.75, 95%: 0.64–0.85) was much higher than that obtained from studies with <180 days follow-up (0.51, 95%: 0.41–0.61). No significant heterogeneity of sensitivity and specificity was identified across studies categorized by sample size, average age, study outcome and pre-specified cut-off value of copeptin.

## Publication bias

The Deek's funnel plot asymmetry test was not statistically significant ($P = 0.44$), which suggested that there is no evident indication of publication bias in the included studies (Fig 5).

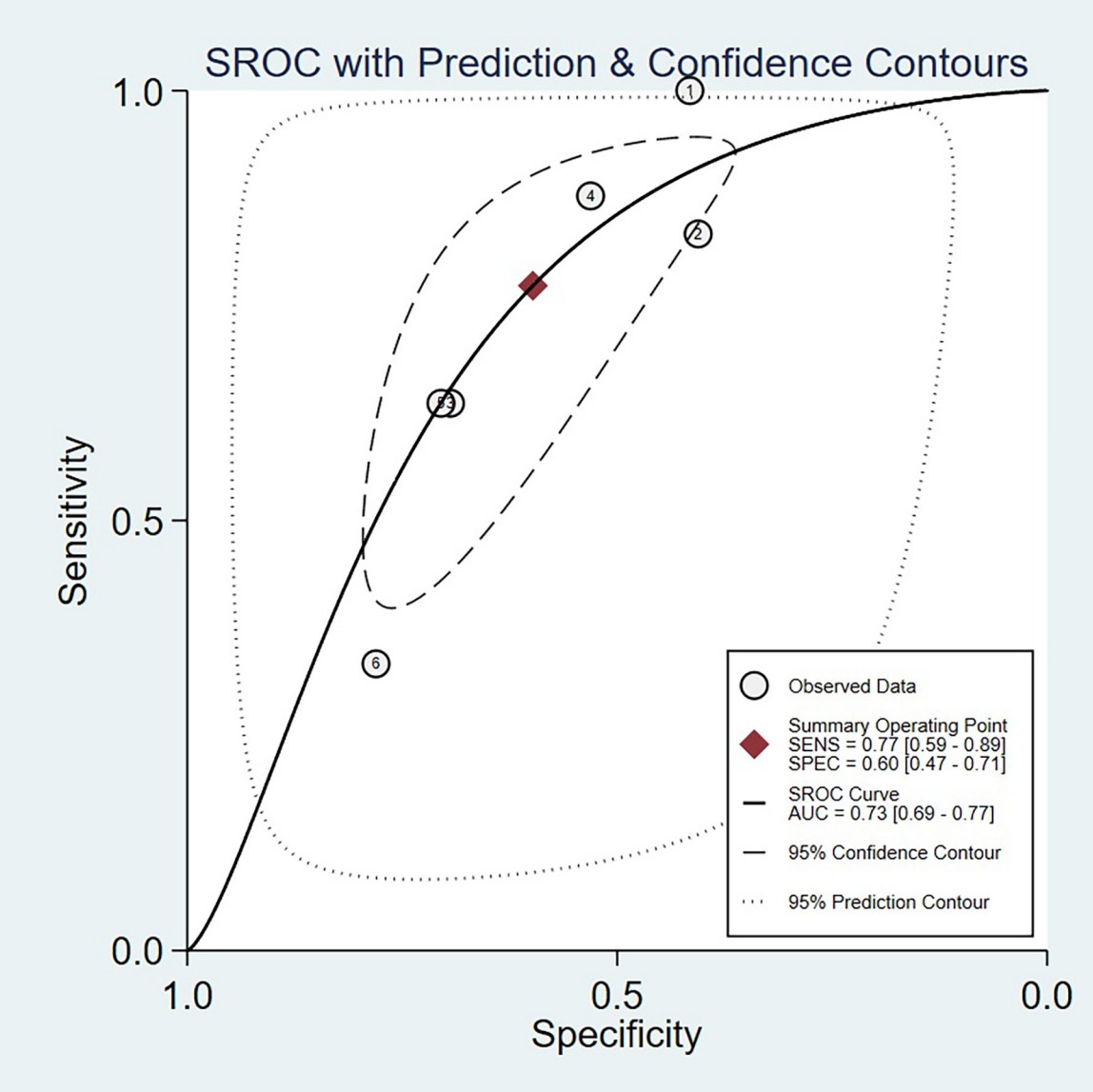

**Fig 3. Summary receiver operating characteristic curve of copeptin in predicting mortality in patients with acute coronary syndrome.**

## Discussion

To our knowledge, this is the first meta-analysis to evaluate the prognostic value of copeptin in patients with ACS. Our meta-analysis including 2269 patients from six studies showed that elevated copeptin was associated with the higher risk of mortality in patients with ACS. The pooled sensitivity and specificity was 0.77 (95% CI: 0.59–0.89) and 0.60 (95% CI: 0.47–0.71), respectively. The AUC of copeptin was 0.73 (95% CI: 0.69–0.77), indicating that copeptin has acceptable performance for predicting mortality in patients with ACS.

Similar to present study, previous studies showed copeptin had prognostic value in patients with heart failure or acute ischemic stroke. Zhong et al. [22] conducted a meta-analysis and reported that elevated copeptin level was associated with an increased risk of all-cause mortality in patients with heart failure (relative risk = 2.64, 95% CI: 2.09–3.32), and the performance

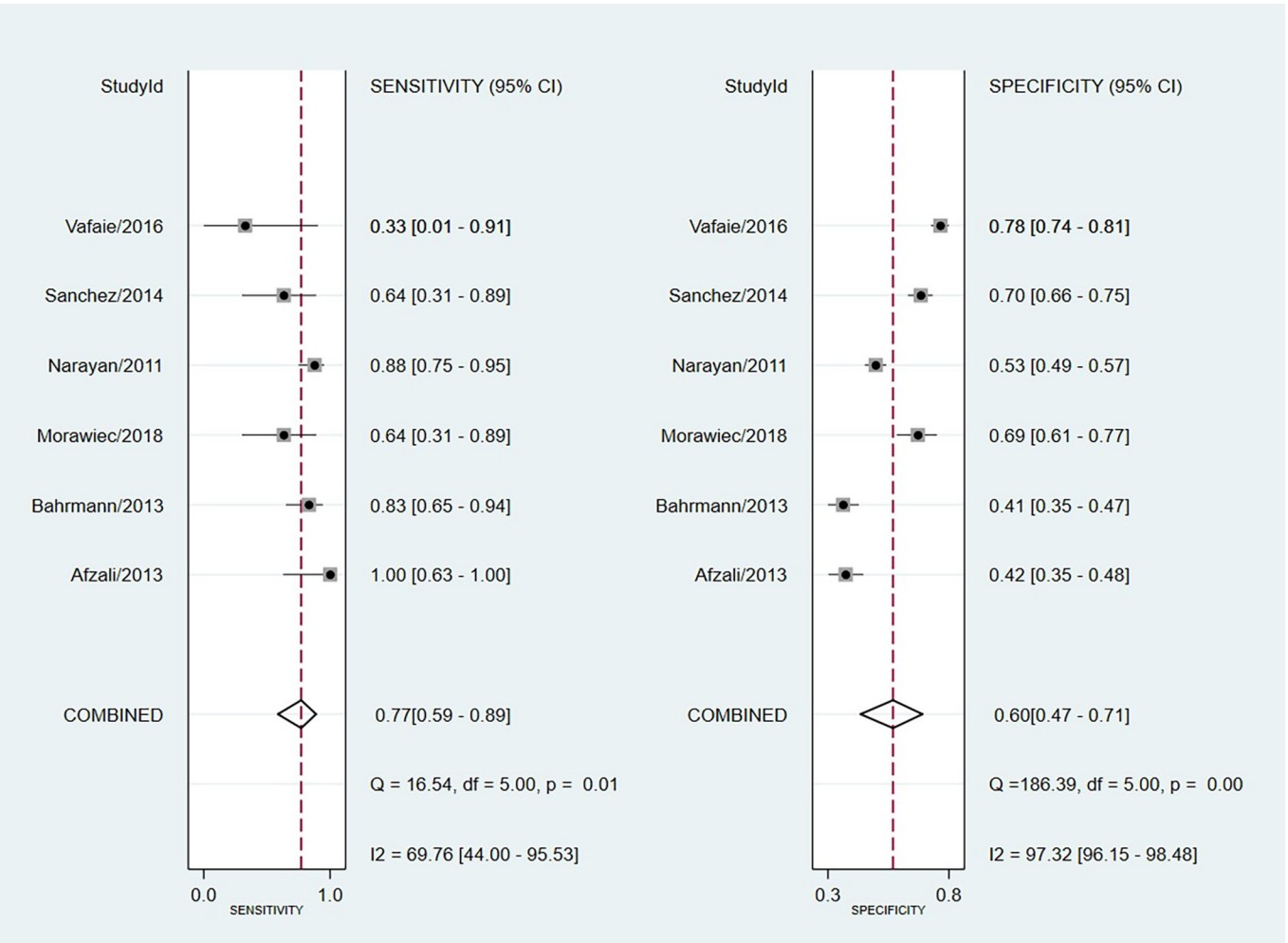

**Fig 4. Forest plot of the sensitivity and specificity of copeptin in predicting mortality in patients with acute coronary syndrome.**

for predicting mortality in patients with heart failure was acceptable (AUC: 0.70, 95% CI: 0.66–0.74). In another meta-analysis based on data from 1976 patients in six studies, it indicated that elevation in plasma copeptin level carried a higher risk of all-cause mortality in patients with acute ischemic stroke (odds ratio = 4.16, 95% CI: 2.77–6.25) [23].

At the current state of knowledge, copeptin could be a potential prognostic marker for ACS due to its acceptable predictive performance. However, there are still several weaknesses for copeptin as prognostic marker for ACS. First, the relationship of copeptin with the risk of

Table 2. Sensitivity analysis of included studies.

| Study omitted | Diagnostic odds ratios | 95% confidence interval |
|---|---|---|
| Afzali, et al. 2013 | 4.76 | 2.84–7.97 |
| Bahrmann, et al. 2013 | 5.39 | 2.95–9.85 |
| Morawiec, et al. 2018 | 4.93 | 2.81–8.66 |
| Narayan, et al. 2011 | 3.57 | 1.88–6.77 |
| Sanchez, et al. 2014 | 4.89 | 2.78–8.61 |
| Vafaie, et al. 2016 | 4.99 | 2.95–8.46 |

**Table 3. Meta-regression analysis to examine the sources of potential heterogeneity of sensitivity and specificity.**

| Parameter | Categories | No. of studies | Sensitivity (95% CI) | P value | Specificity (95% CI) | P value |
|---|---|---|---|---|---|---|
| Study design | Prospective cohort study | 5 | 0.82 (0.71–0.93) | 0.05 | 0.55 (0.45–0.66) | 0.03 |
| | Randomized control trial | 1 | 0.33 (-0.26–0.93) | | 0.78 (0.62–0.94) | |
| No. of study center | Multicenter | 2 | 0.54 (0.23–0.85) | 0.03 | 0.75 (0.64–0.85) | 0.35 |
| | Single-center | 4 | 0.85 (0.75–0.94) | | 0.51 (0.41–0.61) | |
| Sample size | <500 | 4 | 0.74 (0.54–0.95) | 0.60 | 0.62 (0.47–0.76) | 0.97 |
| | ≥500 | 2 | 0.84 (0.60–1.00) | | 0.56 (0.34–0.77) | |
| Average age, years | <70 | 3 | 0.73 (0.45–1.00) | 0.70 | 0.64 (0.48–0.80) | 0.33 |
| | ≥70 | 3 | 0.81 (0.65–0.98) | | 0.55 (0.39–0.72) | |
| Duration of follow-up, days | <180 | 4 | 0.85 (0.75–0.94) | 0.13 | 0.51 (0.41–0.61) | <0.001 |
| | ≥180 | 2 | 0.54 (0.23–0.85) | | 0.75 (0.64–0.85) | |
| Study outcome | All-cause mortality | 4 | 0.79 (0.61–0.98) | 0.80 | 0.62 (0.48–0.76) | 0.91 |
| | Cardiovascular mortality | 2 | 0.74 (0.45–1.00) | | 0.55 (0.34–0.76) | |
| Cut-off value of copeptin | Pre-specified | 5 | 0.72 (0.54–0.90) | 0.28 | 0.61 (0.48–0.74) | 0.79 |
| | Not pre-specified | 1 | 0.88 (0.70–1.00) | | 0.53 (0.23–0.83) | |

CI: confidence interval.

mortality in patients with ACS may be influenced by concomitant diseases, such as stroke, renal failure or sepsis, which will affect its application in clinical practice [9, 24]. Second, current evidences don't support that copeptin is a superior predictor of mortality in patients with ACS compared with cardiac troponin T (cTnT) and NT-proBNP, which are well-established biomarkers for diagnosis and prognosis of ACS [25, 26]. Bahrmann et al. found that high-sensitivity cTnT ≥ 0.014 μg/L alone was significantly associated with cardiovascular mortality, while the net reclassification improvement for cardiovascular mortality was not significant ($P = 0.809$) when copeptin ≥ 14 pmol/L was added [14]. Another study using data from the MERLIN-TIMI 36 Trial reported that copeptin significantly improved the prognostic value when added to traditional clinical factors (net reclassification index = 0.25, $P<0.001$), but copeptin and BNP showed similar discrimination capability (c-statistic: 0.69 for copeptin and 0.69 for BNP) [12]. Similar results were also reported in another hospital-based prospective cohort study [27]. Given that several biomarkers reflect different pathophysiological pathways of response post-MI, we speculated an additive predictive value for combining biomarkers including those of myonecrosis, myocardial strain or stress, and vascular inflammation in CAD prognosis. Previous studies used the multimarker strategy to investigate the risk prediction tool for prognosis of ACS [19, 28, 29], but the prognostic value of copeptin added to other biomarkers in patients with ACS has not been reported.

There are some potential limitations in this study. First, the number of included studies is relatively small. The Deek's funnel plot asymmetry test was not statistically significant, but publication bias may still exist. Second, the cut-off value of copeptin varied across different studies, and we could not determine the optimized cut-off value due to the lack of raw data. It's worthy to explore the optimized cut-off value of copeptin in future studies. Third, most of included studies were conducted in patients with suspected ACS, as a result, we cannot further evaluate the predictive value of copeptin in patients with non-ST-elevation myocardial infarction and ST-elevation myocardial infarction, respectively in meta-regression analysis.

In conclusion, our study shows that copeptin has prognostic value for mortality in patients with ACS. Future studies based on multimarker strategy are needed to evaluate the prognostic value of copeptin in conjunction with other well-established biomarkers for risk stratification in patients with ACS.

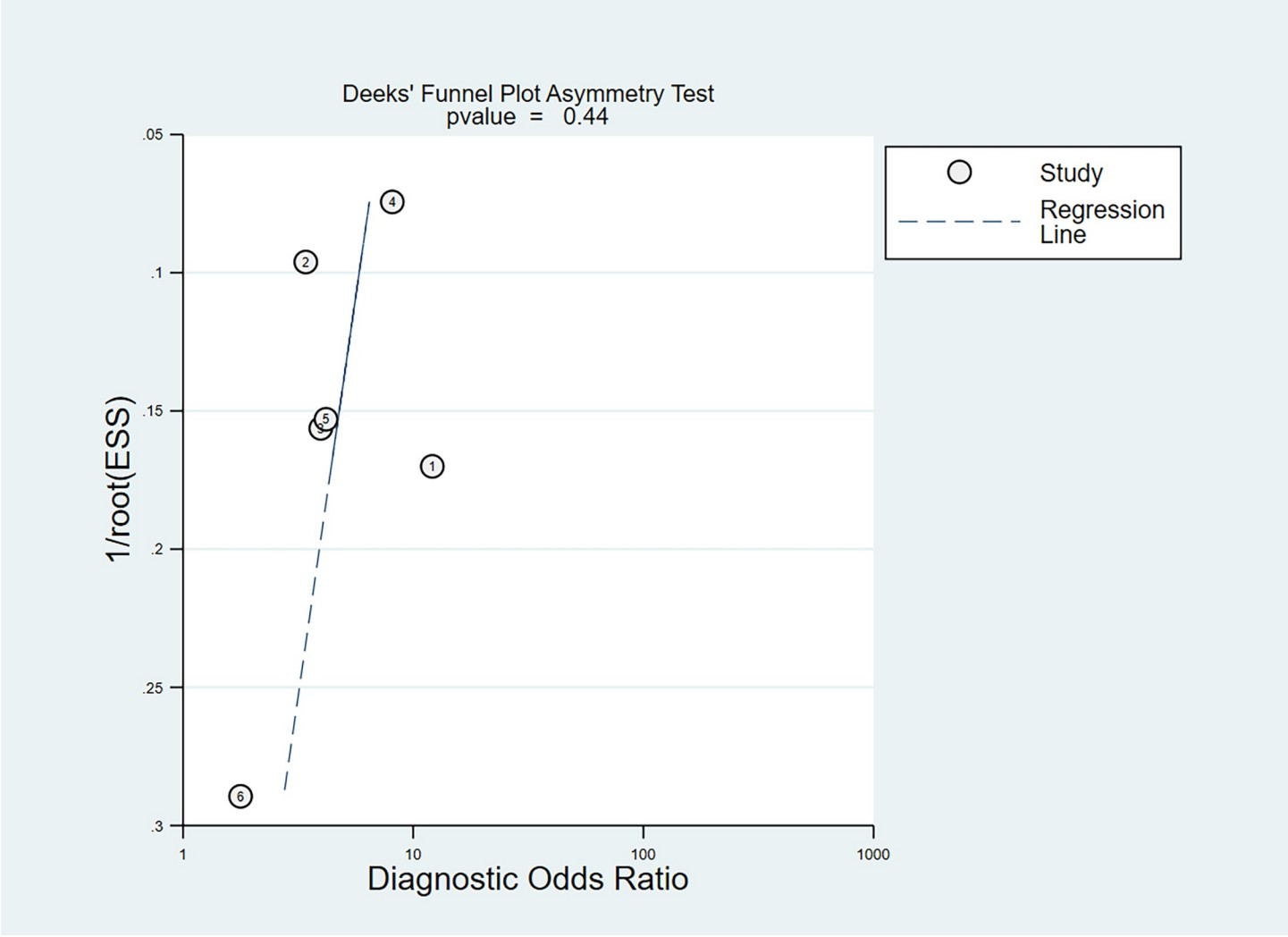

**Fig 5. Funnel plot for detecting publication bias.**

## Supporting information

**S1 File. PRISMA 2019 checklist.**
(PDF)

## Author Contributions

**Conceptualization:** Jiapeng Lu.

**Data curation:** Jiapeng Lu, Siming Wang, Guangda He.

**Formal analysis:** Jiapeng Lu.

**Funding acquisition:** Jiapeng Lu.

**Investigation:** Siming Wang, Guangda He, Yanping Wang.

**Methodology:** Jiapeng Lu.

**Writing – original draft:** Jiapeng Lu.

**Writing – review & editing:** Siming Wang, Guangda He, Yanping Wang.

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
