## [Decision Letter · Decision Letter 0]

19 Jun 2020

PONE-D-20-14294

Prognostic value of copeptin in patients with acute coronary syndrome: a systematic review and meta-analysis

PLOS ONE

Dear Dr. Lu,

Thank you for submitting your manuscript to PLOS ONE. After careful consideration, we feel that it has merit but does not fully meet PLOS ONE’s publication criteria as it currently stands. Therefore, we invite you to submit a revised version of the manuscript that addresses the points raised during the review process.

All issues raised by reviewers are required. Moreover, the authors should add a paragraph describing the limitations of their analysis. Finally, they should better highlight strengths and weaknesses of copeptin as prognostic marker at this stage of knowledge.

We look forward to receiving your revised manuscript.

Kind regards,

Vincenzo Lionetti, M.D., PhD

Academic Editor

PLOS ONE

Journal Requirements:

2. In your Results section, please show the results of the publication bias assessment, including a figure showing the funnel plot.

Reviewers' comments:

Reviewer's Responses to Questions

**Comments to the Author**

1. Is the manuscript technically sound, and do the data support the conclusions?

Reviewer #1: Yes

Reviewer #2: Yes

Reviewer #3: Yes

2. Has the statistical analysis been performed appropriately and rigorously? 

Reviewer #1: Yes

Reviewer #2: Yes

Reviewer #3: Yes

3. Have the authors made all data underlying the findings in their manuscript fully available?

Reviewer #1: Yes

Reviewer #2: Yes

Reviewer #3: Yes

4. Is the manuscript presented in an intelligible fashion and written in standard English?

Reviewer #1: Yes

Reviewer #2: Yes

Reviewer #3: No

5. Review Comments to the Author

Reviewer #1: Manuscript Number: PONE-D-20-14294

Jiapeng Lu et co-authors in the paper submitted to PLOSONE (Prognostic value of copeptin in patients with acute coronary syndrome: a systematic review and meta-analysis) evaluated the value of copeptin in predicting mortality in patients with acute coronary syndrome

The manuscript is very interesting and a good review of the literature has been carried out.

Reviewer #2: Interesting and well conducted meta analysis.

Asbtract; it should be added if we are dealing with in hospital or long term mortality

Abstract authors speak about promising results. An AUC of 0.73 for an exam which is not of routine and has come costs is hard to be defined promising.

Methods/Results: it is not clear if authors used fixed or random effect

Methods/results: meta-regression for age and kind of ACS (stemi vsnstemi) should be performed

Discusson; reduced accuracy of risk scores have been described. this should be commented on (PMID: 32438488)

Reviewer #3: This is a meta-analysis of six studies investigating the prognostic role of copeptin in patients with ACS. All studies included demonstrated that copeptin could significantly discriminate between patients who died and those who did not die during follow-up. Being this a study-level meta-analysis, it adds very little to the available literature.

6. PLOS authors have the option to publish the peer review history of their article (what does this mean?). If published, this will include your full peer review and any attached files.

Reviewer #1: No

Reviewer #2: Yes: Fabrizio D'Ascenzo

Reviewer #3: No

---

## [Author Response · Author response to Decision Letter 0]

7 Jul 2020

EDITOR'S COMMENTS:

1. All issues raised by reviewers are required. Moreover, the authors should add a paragraph describing the limitations of their analysis. Finally, they should better highlight strengths and weaknesses of copeptin as prognostic marker at this stage of knowledge.

Response: We had revised the manuscript according to reviewers’ comments and provided responses to each comment. We had a paragraph to describe the potential limitations of our study on page 10, lines 235-243 as follows:

There are some potential limitations in this study. First, the number of included studies is relatively small. The Deek’s funnel plot asymmetry test was not statistically significant, but publication bias may still exist. Second, the cut-off value of copeptin varied across different studies, and we could not determine the optimized cut-off value due to the lack of raw data. It’s worthy to explore the optimized cut-off value of copeptin in future studies. Third, most of included studies were conducted in patients with suspected ACS, as a result, we cannot further evaluate the predictive value of copeptin in patients with non-ST-elevation myocardial infarction and ST-elevation myocardial infarction, respectively in meta-regression analysis.

According to this comment, we had highlighted strengths and weaknesses of copeptin as prognostic marker based on the discussions in previous version of manuscript on page 10, lines 211-234 as follows:

At the current state of knowledge, copeptin could be a potential prognostic marker for ACS due to its acceptable predictive performance. However, there are still several weaknesses for copeptin as prognostic marker for ACS. First, the relationship of copeptin with the risk of mortality in patients with ACS may be influenced by concomitant diseases, such as stroke, renal failure or sepsis, which will affect its application in clinical practice [1, 2]. Second, current evidences don’t support that copeptin is a superior predictor of mortality in patients with ACS compared with cardiac troponin T (cTnT) and NT-proBNP, which are well-established biomarkers for diagnosis and prognosis of ACS [3, 4]. Bahrmann et al. found that high-sensitivity cTnT ≥ 0.014 µg/L alone was significantly associated with cardiovascular mortality, while the net reclassification improvement for cardiovascular mortality was not significant (P = 0.809) when copeptin ≥ 14 pmol/L was added [5]. Another study using data from the MERLIN-TIMI 36 Trial reported that copeptin significantly improved the prognostic value when added to traditional clinical factors (net reclassification index=0.25, P<0.001), but copeptin and BNP showed similar discrimination capability (c-statistic: 0.69 for copeptin and 0.69 for BNP) [6]. Similar results were also reported in another hospital-based prospective cohort study [7]. Given that several biomarkers reflect different pathophysiological pathways of response post-MI, we speculated an additive predictive value for combining biomarkers including those of myonecrosis, myocardial strain or stress, and vascular inflammation in CAD prognosis. Previous studies used the multimarker strategy to investigate the risk prediction tool for prognosis of ACS[8-10], but the prognostic value of copeptin added to other biomarkers in patients with ACS has not been reported.

2. Please ensure that your manuscript meets PLOS ONE's style requirements, including those for file naming. The PLOS ONE style templates can be found at https://journals.plos.org/plosone/s/file?id=wjVg/PLOSOne_formatting_sample_main_body.pdf and https://journals.plos.org/plosone/s/file?id=ba62/PLOSOne_formatting_sample_title_authors_affiliations.pdf

Response: We have updated our manuscript according to PLOS ONE’s style requirements.

3. In your Results section, please show the results of the publication bias assessment, including a figure showing the funnel plot.

Response: The results of the publication bias were included in the first submitted manuscript on page 9, lines 188-190 as follows:

The Deek’s funnel plot asymmetry test was not statistically significant (P=0.44), which suggested that there is no evident indication of publication bias in the included studies (Fig 5).

According to this comment, we have added a funnel plot for detecting publication bias as Fig 5.

4. PLOS requires an ORCID iD for the corresponding author in Editorial Manager on papers submitted after December 6th, 2016. Please ensure that you have an ORCID iD and that it is validated in Editorial Manager. To do this, go to ‘Update my Information’ (in the upper left-hand corner of the main menu), and click on the Fetch/Validate link next to the ORCID field. This will take you to the ORCID site and allow you to create a new iD or authenticate a pre-existing iD in Editorial Manager. Please see the following video for instructions on linking an ORCID iD to your Editorial Manager account: https://www.youtube.com/watch?v=_xcclfuvtxQ.

Response: The corresponding author’s ORCID is 0000-0001-9179-4587. We have added it in the Editorial Manager system.

Response: No supporting information files was provided in our study.

REVIEWERS' COMMENTS:

Reviewer #1: 

1. Manuscript Number: PONE-D-20-14294. Jiapeng Lu et co-authors in the paper submitted to PLOSONE (Prognostic value of copeptin in patients with acute coronary syndrome: a systematic review and meta-analysis) evaluated the value of copeptin in predicting mortality in patients with acute coronary syndrome. The manuscript is very interesting and a good review of the literature has been carried out.

Response: Thank you for your comments. 

Reviewer #2: 

1. Interesting and well conducted meta analysis. Asbtract; it should be added if we are dealing with in hospital or long term mortality

Response: Thank you for your comments. according to this comment, we updated the descriptions in the Abstract on page 1, lines 18-20 as follows:

Background: The aim of this study was to evaluate the value of copeptin in predicting mortality including both short-term and long-term mortality in patients with acute coronary syndrome (ACS).

2. Abstract authors speak about promising results. An AUC of 0.73 for an exam which is not of routine and has come costs is hard to be defined promising.

Response: We have updated the word we used in the Abstract on page 1, lines 36-37 as follows: 

Copeptin has acceptable prognostic value for mortality in patients with ACS.

3. Methods/Results: it is not clear if authors used fixed or random effect

Response: We mentioned that we used a random-effects model to calculate pooled sensitivity and specificity in Methods section on page 5, lines 97 as follows:

We calculated pooled sensitivity and specificity using a random-effects model.

4. Methods/results: meta-regression for age and kind of ACS (stemi vs nstemi) should be performed

Response: According to this comment, we have added the results of meta-regression for age in Table 3 as follows:

Table 3. Meta-regression analysis to examine the sources of potential heterogeneity of sensitivity and specificity

Parameter Categories No. of studies Sensitivity (95% CI) P value Specificity (95% CI) P value

Study design Prospective cohort study 5 0.82 (0.71 - 0.93) 0.05 0.55 (0.45 - 0.66) 0.03

 Randomized control trial 1 0.33 (-0.26 - 0.93) 0.78 (0.62 - 0.94) 

No. of study center Multicenter 2 0.54 (0.23 - 0.85) 0.03 0.75 (0.64 - 0.85) 0.35

 Single-center 4 0.85 (0.75 - 0.94) 0.51 (0.41 - 0.61) 

Sample size <500 4 0.74 (0.54 - 0.95) 0.60 0.62 (0.47 - 0.76) 0.97

 ≥500 2 0.84 (0.60 - 1.00) 0.56 (0.34 - 0.77) 

Average age, years <70 3 0.73 (0.45 – 1.00) 0.70 0.64 (0.48 - 0.80) 0.33

 ≥70 3 0.81 (0.65 – 0.98) 0.55 (0.39 - 0.72) 

Duration of follow-up, days <180 4 0.85 (0.75 - 0.94) 0.13 0.51 (0.41 - 0.61) <0.001

 ≥180 2 0.54 (0.23 - 0.85) 0.75 (0.64 - 0.85) 

Study outcome All-cause mortality 4 0.79 (0.61 - 0.98) 0.80 0.62 (0.48 - 0.76) 0.91

 Cardiovascular mortality 2 0.74 (0.45 - 1.00) 0.55 (0.34 - 0.76) 

Cut-off value of copeptin Pre-specified 5 0.72 (0.54 - 0.90) 0.28 0.61 (0.48 - 0.74) 0.79

 Not pre-specified 1 0.88 (0.70 - 1.00) 0.53 (0.23 - 0.83) 

CI: confidence interval.

We also updated the results on page 8, lines 179-181 as follows:

No significant heterogeneity of sensitivity and specificity was identified across studies categorized by sample size, average age, study outcome and cut-off value of copeptin.

Most of included studies recruited patients with suspected ACS including both STEMI and NSTEMI. But original data used to calculate sensitivity and specificity were not available for STEMI and NSTEMI, respectively. Thus, we cannot conduct meta-regression analysis for the kind of ACS. 

5. Discusson; reduced accuracy of risk scores have been described. this should be commented on (PMID: 32438488)

Response: We didn’t mention about the reduced accuracy of risk scores in the Discussion section. And the publication (PMID: 32438488) you mentioned is about the accuracy of the PARIS score and PCI complexity to predict ischemic events in patients treated with very thin stents in unprotected left main or coronary bifurcations. We realized that this study is not relevant to the research top in our study. Therefore, we didn’t make any change based on this comment.

Reviewer #3: 

1. This is a meta-analysis of six studies investigating the prognostic role of copeptin in patients with ACS. All studies included demonstrated that copeptin could significantly discriminate between patients who died and those who did not die during follow-up. Being this a study-level meta-analysis, it adds very little to the available literature.

Response: Thank you for your comments. Among the six included studies, there is one study reporting no significant association of copeptin with the risk of mortality among patients with ACS[11]. In addition, our study was not only to evaluate the association between copeptin and mortality, but also to quantify the value of copeptin in predicting mortality in patients with ACS.

References:

1. Morgenthaler NG, Struck J, Jochberger S, Dunser MW. Copeptin: clinical use of a new biomarker. Trends in endocrinology and metabolism: TEM. 2008;19(2):43-9.

2. Katan M, Fluri F, Morgenthaler NG, Schuetz P, Zweifel C, Bingisser R, et al. Copeptin: a novel, independent prognostic marker in patients with ischemic stroke. Annals of neurology. 2009;66(6):799-808.

3. Katus HA, Remppis A, Neumann FJ, Scheffold T, Diederich KW, Vinar G, et al. Diagnostic efficiency of troponin T measurements in acute myocardial infarction. Circulation. 1991;83(3):902-12.

4. Omland T, Persson A, Ng L, O'Brien R, Karlsson T, Herlitz J, et al. N-terminal pro-B-type natriuretic peptide and long-term mortality in acute coronary syndromes. Circulation. 2002;106(23):2913-8.

5. Bahrmann P, Bahrmann A, Breithardt OA, Daniel WG, Christ M, Sieber CC, et al. Additional diagnostic and prognostic value of copeptin ultra-sensitive for diagnosis of non-ST-elevation myocardial infarction in older patients presenting to the emergency department. Clinical chemistry and laboratory medicine. 2013;51(6):1307-19.

6. O'Malley RG, Bonaca MP, Scirica BM, Murphy SA, Jarolim P, Sabatine MS, et al. Prognostic performance of multiple biomarkers in patients with non-ST-segment elevation acute coronary syndrome: analysis from the MERLIN-TIMI 36 trial (Metabolic Efficiency With Ranolazine for Less Ischemia in Non-ST-Elevation Acute Coronary Syndromes-Thrombolysis In Myocardial Infarction 36). Journal of the American College of Cardiology. 2014;63(16):1644-53.

7. Khan SQ, Dhillon OS, O'Brien RJ, Struck J, Quinn PA, Morgenthaler NG, et al. C-terminal provasopressin (copeptin) as a novel and prognostic marker in acute myocardial infarction: Leicester Acute Myocardial Infarction Peptide (LAMP) study. Circulation. 2007;115(16):2103-10.

8. Sabatine MS, Morrow DA, de Lemos JA, Gibson CM, Murphy SA, Rifai N, et al. Multimarker approach to risk stratification in non-ST elevation acute coronary syndromes: simultaneous assessment of troponin I, C-reactive protein, and B-type natriuretic peptide. Circulation. 2002;105(15):1760-3.

9. O'Donoghue ML, Morrow DA, Cannon CP, Jarolim P, Desai NR, Sherwood MW, et al. Multimarker Risk Stratification in Patients With Acute Myocardial Infarction. Journal of the American Heart Association. 2016;5(5).

10. Schnabel RB, Schulz A, Messow CM, Lubos E, Wild PS, Zeller T, et al. Multiple marker approach to risk stratification in patients with stable coronary artery disease. European heart journal. 2010;31(24):3024-31.

11. Sanchez M, Llorens P, Herrero P, Martin-Sanchez FJ, Pinera P, Miro O, et al. The utility of copeptin in the emergency department as a predictor of adverse outcomes in non-ST-elevation acute coronary syndrome: the COPED-PAO study. Emergency medicine journal : EMJ. 2014;31(4):286-91.

---

## [Decision Letter · Decision Letter 1]

14 Aug 2020

Prognostic value of copeptin in patients with acute coronary syndrome: a systematic review and meta-analysis

PONE-D-20-14294R1

Dear Dr. Lu,

We’re pleased to inform you that your manuscript has been judged scientifically suitable for publication and will be formally accepted for publication once it meets all outstanding technical requirements.

Kind regards,

Vincenzo Lionetti, M.D., PhD

Academic Editor

PLOS ONE

Additional Editor Comments (optional):

Reviewers' comments:

Reviewer's Responses to Questions

**Comments to the Author**

1. If the authors have adequately addressed your comments raised in a previous round of review and you feel that this manuscript is now acceptable for publication, you may indicate that here to bypass the “Comments to the Author” section, enter your conflict of interest statement in the “Confidential to Editor” section, and submit your "Accept" recommendation.

Reviewer #2: All comments have been addressed

Reviewer #3: (No Response)

Reviewer #4: All comments have been addressed

2. Is the manuscript technically sound, and do the data support the conclusions?

Reviewer #2: (No Response)

Reviewer #3: Partly

Reviewer #4: Yes

3. Has the statistical analysis been performed appropriately and rigorously? 

Reviewer #2: (No Response)

Reviewer #3: Yes

Reviewer #4: Yes

4. Have the authors made all data underlying the findings in their manuscript fully available?

Reviewer #2: (No Response)

Reviewer #3: Yes

Reviewer #4: Yes

5. Is the manuscript presented in an intelligible fashion and written in standard English?

Reviewer #2: (No Response)

Reviewer #3: Yes

Reviewer #4: Yes

6. Review Comments to the Author

Reviewer #2: (No Response)

Reviewer #3: My concerns regarding the utility and novelty of a study-level meta-analysis of studies yielding similar results still remain.

In the rebuttal letter, the authors state that "there is one study reporting no significant association of copeptin with the risk of mortality among patients with ACS" (Sanchez M, et al. Emergency

medicine journal : EMJ. 2014;31(4):286-91.). In the latter study of patients with NSTEACS, copeptin was associated with risk of death during follow-up, however this association disappeared after adjusting by baseline features or troponin level. This point has not been addressed in the present meta-analysis, as only the impact of age was analyzed in a meta-regression analysis. In fact, only crude data have been used, which are totally in line with all other studies.

The study limitations continue to outweigh the strengths of the paper.

Reviewer #4: (No Response)

7. PLOS authors have the option to publish the peer review history of their article (what does this mean?). If published, this will include your full peer review and any attached files.

Reviewer #2: **Yes: **Fabrizio D'Ascenzo

Reviewer #3: No

Reviewer #4: No

---

## [Editor Report · Acceptance letter]

19 Aug 2020

PONE-D-20-14294R1 

Prognostic value of copeptin in patients with acute coronary syndrome: a systematic review and meta-analysis 

Dear Dr. Lu:

I'm pleased to inform you that your manuscript has been deemed suitable for publication in PLOS ONE. Congratulations! Your manuscript is now with our production department. 

Kind regards, 

on behalf of

Prof. Vincenzo Lionetti 

Academic Editor

PLOS ONE